# Features of Protein Unfolding Transitions and Their Relation to Domain Topology Probed by Single-Molecule FRET

**DOI:** 10.3390/biom13091280

**Published:** 2023-08-22

**Authors:** Nuno Bustorff, Jörg Fitter

**Affiliations:** 1ER-C-3 Structural Biology & IBI-6 Cellular Structural Biology, Forschungszentrum Jülich, 52425 Jülich, Germany; n.bustorff@fz-juelich.de; 2AG Biophysik, I. Physikalisches Institut (IA), RWTH Aachen University, 52074 Aachen, Germany

**Keywords:** protein folding, single-molecule Förster resonance energy transfer, domain topology, unfolding/folding-induced conformational changes, molten globule intermediate state

## Abstract

A protein fold is defined as a structural arrangement of a secondary structure in a three-dimensional space. It would be interesting to know whether a particular fold can be assigned to certain features of the corresponding folding/unfolding transitions. To understand the underlying principles of the manifold folding transitions in more detail, single-molecule FRET is the method of choice. Taking the two-domain protein phosphoglycerate kinase (PGK) as an example, we investigated denaturant-induced unfolded states of PGK using the above method. For this purpose, different intramolecular distances within the two domains were measured. In addition to the known two-state transition, a transition with a compact folding intermediate was also identified in each of the two domains. Based on the structural homology of the domains (characterized by a Rossmann fold) and the striking similarity in the features of the measured distance changes during unfolding, clear evidence emerged that the underlying domain topology plays an important role in determining the observed structural changes.

## 1. Introduction

When analyzing the folding of proteins, fluorescent dyes are typically attached at different positions in the protein so that single-molecule Förster resonance energy transfer (smFRET) can be used to measure intramolecular distances during folding/unfolding transitions [1,2,3]. Especially in the case of multidomain proteins, these distances can be chosen to be between or within individual domains. Both options provide valuable information and can complement each other [4]. In practice, the choice of dye labeling positions is limited. On the one hand, the typical replacement of the existing amino acid at the position by the cysteines (to which the dyes are then bound) must not change the protein structure. On the other hand, the distance between the two positions must be chosen so that it approximately corresponds to the Förster radius, so that the method is sensitive enough to accurately measure even small distance changes during folding/unfolding transitions [5,6,7]. In the recent past, several studies have shown how different intramolecular distances in one and the same protein change quite differently during the unfolding transition [4,8,9,10]. Such an approach allows more detailed insights into the complex behavior of folding/unfolding transitions, e.g., the identification of folding/unfolding intermediates. In any case, it can be assumed that the topology of the protein or of the domain also contributes to the individual properties of the unfolding-induced distance changes.

Taking the two-domain protein phosphoglycerate kinase from yeast (yPGK) as an example, we aim to investigate the properties of distance changes within each domain under denaturing conditions. The PGK enzyme (415 residues) is composed of two structurally homologous domains, the N- and the C-terminal domains (Figure 1).

The topology, i.e., the sequential order of the secondary structure elements and their spatial arrangement, is nearly identical for both domains and displays a Rossmann-like α/β/α sandwich fold [11,12]. As indicated in earlier studies, the individual domains represent independent folding units, since they can fold into native-like structures, even as isolated domains (i.e., not being part of full-length PGK) [13,14]. We have recently performed a study, where we measured the evolution of intra-domain and inter-domain distances as a function of chemical denaturant concentration [4]. In that study, mainly intra-domain distances within the C-terminal domain were investigated and discussed. 

Here, we prepared additional cysteine double mutants to measure corresponding denaturant-induced intra-domain distance changes within the N-terminal domain. Using such a more extended data set (i.e., data from two homologous domains within one protein) from smFRET measurements, the goal was to identify regularities in the distance changes and understand them with respect to the underlying topology.

## 2. Materials and Methods

### 2.1. Sample Preparation

The cysteine double mutants of the gene of phosphoglycerate kinase from *Saccaharomyces cerevisea* were produced as described in [15]. In addition to the replacement of the natural C97 with a serine, a pair of cysteine residues was incorporated using site-directed mutagenesis at the corresponding positions, here at S1C-G88C and at T34-Q135C. All yPGK variants were expressed in *E. coli* using BL21(DE3) competent cells (New England Biolabs, Frankfurt, Germany, cat. no. C2527H). DE3 lysogen expresses T7 RNA polymerase (RNAP) from the bacterial genome under the control of the lac repressor. Cells were grown at 37 °C in Terrific Broth medium (TB) (MP Biomedicals, Irvine, CA, USA cat. no. 113046022-CF) with shaking from OD_600nm_ = 0.2 up to OD_600nm_ = 1. Afterwards, protein expression was induced by addition of 1 mM Isopropyl s-D-1-thiogalactopyranoside (IPTG), and incubation was continued for 3 h at the same temperature. Finally, protein expression was stopped by harvesting cells at 6000× *g* for 15 min at 4 °C. The cell pellet was resuspended in lysis buffer (30 mM MOPS with pH of 7.5, 500 mM NaCl, 0.001 mM TCEP) supplemented with cOmplete™ Protease Inhibitor Cocktail (1 tablet/50 mL) (Roche, Basel, Switzerland cat. no. 11697498001). Cells were opened using a cell disruptor by passing them 2 times at a pressure of 1.7 mbar to increase protein recovery. Cell lysate was centrifuged at 30,000× *g* for 45 min to remove cell debris. A salting-out step using Ammonium sulphate precipitation followed. First, the clear lysate was mixed with 40% (*w*/*v*) (NH_4_)_2_SO_4_ (AppliChem GmbH, Darmstadt, Germany cat. no. 141140) for 1 h at 4 °C. Non-precipitated proteins were recovered after centrifuging the solution at 5000× *g* for 45 min. The process was repeated with 60% (*w*/*v*) of (NH_4_)_2_SO_4_. Soluble proteins were further purified with Nickel-Nitriloacetic acid (Ni-NTA) affinity chromatography. Batch purification was performed using 5 mL of resin from a 50% (*w*/*v*) suspension of ProtinoR Ni-NTA agarose (Macherey-Nagel, Düren Germany cat. no. 745400.500). Beads were equilibrated in lysis buffer before incubating them for 1 h with the protein solution. Two washes each with 5 column volumes (CVs) using incremental amounts of imidazole in lysis buffer, namely 10 mM and 20 mM, followed. Lastly, the protein of interest (POI) was eluted after passing 5 CVs of lysis buffer supplemented with 250 mM imidazole. The elution was dialyzed overnight in 400× volume excess of lysis buffer plus 1:10 (*w*/*w*) of TEV-Hisx6 protease (produced in house). TEV protease and uncut proteins were removed through reverse Ni-NTA affinity chromatography, as described above, where the protein of interest was recovered in the flow through. Finally, protein preparation was polished in lysis buffer with size-exclusion chromatography using preparative HiLoad 16/600 Superdex 200 pg (Cytiva, Marlborough, MA, USA cat. no. 28989335). The presence of protein in the fractions of interest was confirmed by SDS-PAGE and Western blotting. 

The labeling reaction was performed simultaneously for both positions by mixing equal parts of 10× final molar excess of maleimide dyes (Thermo Scientific™, Waltham, MA, USA Alexa 488 cat. no. A10254 and Alexa 647 cat. no. A20347) with 10 µM yPGK in lysis buffer. Changes in the labeling scheme were used for some PGK variants by mixing different parts of donor and acceptor into the protein solution in order to maximize labeling yield. The reaction was incubated for 2 h in the dark at room temperature and continued overnight at 4 °C. Excess of free dye was removed by passing the labeling mixture through long Zeba™ (Thermo Scientific™, Waltham, MA, USA) dye and biotin-removal 2 mL spin column (Thermo Scientific™, Waltham, MA, USA cat. no. A44298). Sample was kept at 4 °C until used in the confocal microscope.

### 2.2. Confocal Microscopy and smFRET Analysis

Fluorescently labeled samples were measured on a confocal fluorescence microscope Microtime 200 (PicoQuant, Berlin, Germany). First, the confocal volumes of each laser excitation (485 nm and 640 nm) were determined using standard fluorophores (Atto 655-NHS ester or Alexa488-NHS ester) with known diffusion coefficients. For this purpose, 30 µL of the sample solution was pipetted on a small cover slide of 22 × 22 mm with a thickness of 0.17 mm (Marienfeld, Lauda-Königshofen, Germany cat. no. 0107052) on top of a water-immersion high-numerical-aperture objective (UPLanSApo 60×/1.2 W cat. no. N1480800, Olympus, Hamburg, Germany). For single-molecule FRET measurements, 100 µL of 13 pM of fluorescently labeled sample were pipetted on a cover slide sealed with parafilm to reduce sample evaporation. Measurements were performed in PIE mode with a 20 MHz repetition rate [16]. Laser power was adjusted to 9 µW for the 485 nm and 640 nm lasers. Samples were typically measured for approximately 6 h. After every 2 h, the sample was replaced by a fresh one. 

Data from the measurements were analyzed with the PAM toolkit [17]. Further details of the data treatment and the analysis are given in reference [4]. The obtained smFRET histograms were fitted individually with one or two Gaussians, in which the obtained peak positions of the Gaussians corresponded to the mean FRET efficiencies <*E*>. The latter were used to determine the inter-dye distance with
(1)RDA=R0(1−〈E〉〈E〉)16
where *R*_0_ represents the Förster radius which was determined for each measuring condition individually. The obtained values for *R*_0_ ranged between 51.8 and 53.2 Å; for details, see [4]. Three-dimensional protein structures were displayed with PyMOL [18], and topology diagrams were produced with the PDBsum webtool [19]. 

## 3. Results and Discussion

### 3.1. Development of Denaturant-Induced Intra-Domain Distances

In order to extend our understanding of denaturant-induced unfolding transitions in yPGK, we designed new cysteine double mutants for measuring intra-domain distance changes. Since we already analyzed three different distances within the C-terminal domain in a previous work (see Appendix A Appendix A and ref. [4]), here, we focused on further distance changes within the N-terminal domain upon guanidine hydrochloride (GndHCl)-induced unfolding (see Appendix A). In total, we measured a series of six different intra-domain distances at increasing denaturant concentrations at equilibrium conditions in our analysis, three within the N-terminal and three within the C-terminal domain. Surprisingly, we obtained three different types of unfolding transitions from all six distances, for which examples are shown in Figure 2.

First, we observed the already often described two-state transition (TS), where a compact native state was more or less well separated from a less compact unfolded state. Here, the native state was gradually depopulated at the expense of the population of the unfolded state. The second type of transition was characterized by a compact intermediate state (CI). Remarkably, the compacted intermediate had an inter-dye distance that was smaller than that in the native protein structure. Under high denaturant concentrations (>1 M), the compact intermediate state disappeared more or less completely, and only a fully unfolded state remained. Finally, we can report on another unfolding transition which showed only one population with a largely unchanged inter-dye distance during the whole transition, here called “no transition” (NT). Since the C1-variant also unfolded very well during the transition in this case (see ensemble unfolding data in the Appendix A), the behavior observed here indicates that the fully unfolded state (by chance) had the same inter-dye distance as the folded state. At this point, it is important to note that although we observed a different evolution of distances during the unfolding of the different variants, we assumed that in all cases, the respective variants had the same overall folding progression for the total protein. This assumption was also supported by CD-spectroscopy and tryptophan fluorescence data, which showed rather similar unfolding transitions for all variants (see Appendix A).

The results obtained from all six different inter-domain distance analyses are given in Table 1. Here, the corresponding inter-dye distances, R_DA_, for the folded, the fully unfolded, and, if present, the intermediate state are presented. The native R_DA_ –values agreed very well with the distances we obtained between the dye attachment positions based on the yPGK structure (considering the accessible volume approach of attached dyes, see refs. [4,20]). For the two-state and the compact intermediate transitions, we obtained R_DA_-values for the fully unfolded state which were clearly larger than the native R_DA_. This indicated the expected structural expansion of the unfolded state for a globular protein.

To better understand the obtained data regarding the unfolding-induced expansion of the protein structure, we chose an approach in which the numbers of amino acid residues between the dye attachment positions of the corresponding variants were taken into account (see Figure 3). Obviously, the number and the spatial arrangement of amino acids between the attachment positions determined the measured the inter-dye distance. In particular, for the unfolded state, we assumed a random-coil structure of the amino acid chain. In this case, one can estimate the end-to-end distance of this chain using $<R^{2}{>}_{}=n\cdot l^{2}$, where *n* gives the number of residues and *l* is a persistence length [21]. With the exception of the N2- and C3-variants, we saw a good agreement between the calculated $\sqrt{<R^{2}>}$-values and the R_DA_ values for the unfolded state (Table 1). However, a complete match for all variants was maybe unlikely, since not only did the spatial arrangement of the amino acids, which were enclosed by the two labeling attachment positions, determine the measured R_DA_ values but there was also a possible impact of the adjacent structural elements.

The most remarkable aspect of the data obtained is given by the fact that the intermediate state was characterized by an inter-dye distance which was even smaller than that of the native state R_DA_ (see N1- and C2-variants). Here, a transient compacting of the structure during the unfolding transition was assumed. Furthermore, additional refolding studies indicated that the compact intermediate state occurred not only during unfolding but also during refolding. A similar reversible unfolding/refolding behavior was already observed earlier for yPGK between the native and the fully unfolded state [15]. It should be noted here that such a compact intermediate was observed earlier for another α/β protein, namely apoflavodoxin [9]. In this study, the authors assigned the observed intermediate to a molten globule (MG) state. Such MG states are typically thermodynamically stable states and exhibit the presence of secondary-structure elements, but with a less compact arrangement of these elements [22]. Also, in the case of yPGK, a molten globule intermediate was already detected in previous ensemble NMR and fluorescence studies [23,24]. The fact that in our study, a smaller inter-dye distance was measured for the intermediate state than for the native state (visible for the N1- and the C2-variants and, therefore, appearing in both domains, see Table 1) indicates a transient misfolding of the protein during the unfolding/folding transition. 

Finally, it is worth noting that for the three intra-domain distances within each domain, we obtained exactly the three types of unfolding transitions mentioned above. A more detailed discussion of this aspect follows in the subsequent subsection. 

### 3.2. Unfolding Transitions Related to Domain Topology

In order to establish a possible link between the folding schemes found and the structural elements enclosed between the two dye attachment positions, we chose a special point of view of the yPGK structures involved. Here, the enclosed structural elements were highlighted and positioned within the corresponding domain for each variant (Figure 3). A closer look at this arrangement allows the following observations: (i) The variants with the largest numbers of amino acid residues between the dye attachment positions, i.e., N1-variant (1–135) and C2-variant (202–290), exhibited compact intermediate unfolding transitions. (ii) If the number of residues was reduced and one included only the first part (i.e., upstream with respect to the sequential order) of those residues which were enclosed in the N1- and C2-variants, we obtained the N2-variant (1–88) and the C1-variant (202–256). Both of these variants showed an unfolding transition which was classified by the “no transition” type. (iii) Another reduction in the number of residues, again with respect to those of the N1-and C2-variants, but including the second part (i.e., downstream with respect to the sequential order) obtained the N3-variant (34–135) and the C3-variant (256–290). Again, these two variants also exhibited the same type of unfolding transition, this time with two-state unfolding. Such a systematic behavior indicates a special assignment between the different properties of the unfolding transition and the different inter-dye distances measured for the individual variant. 

Essentially, the different inter-dye distances measure structural properties in different parts of the protein and along different spatial directions. We can assume that the macromolecular structure of a protein with thousands of atoms shows a rather complex unfolding behavior on the atomic level. As a consequence, we observed part of this complexity (in terms of three different types of unfolding transitions) in the measured data because we sampled conformational changes with a sensitivity approaching the atomic level, here with smFRET with a spatial resolution in the nanometer to sub-nanometer regimes. The striking similarity in unfolding properties observed for the distances between the dyes within the N-terminal and C-terminal domains can best be understood by taking a closer look at the domain topology (see Figure 4).

As in Figure 3, in the topology plots, we also see the pronounced structural homology of the enclosed structural elements (highlighted in color in Figure 4) for those variants that showed the same type of unfolding transition. Although we must assume that all variants had more or less the same folding/unfolding states at a certain denaturant concentration, different distances within the protein were measured for the individual variants. In addition, from ensemble CD and fluorescence data, we found that secondary structure elements were present in the native state and, to a large extent, in the denaturant regime at which intermediate states appeared (see Appendix A Appendix A). As seen in previous studies, the intermediate states were not visible in the ensemble unfolding transitions, which were also measured under equilibrium conditions. However, comparison of our ensemble data with the corresponding smFRET data showed that the secondary structure elements were almost completely intact when the formation of the molten global intermediate began (at 0.3 M and at 0.5 M for the C3-variant and for the N1-variant, respectively). At a denaturant concentration close to C_1/2_, the intermediate states started to disappear at the expense of the unfolded state. 

Taking these facts into account, a comparison of the folding transition types (Figure 2) with the domain topologies (Figure 4) allows the following observations and explanation attempts.

All enclosed structure elements represent, at least a part, i.e., one to four strands out of the six parallel β-strands, of the Rossmann fold, in most cases sandwiched by the accompanied α-helices.Significant inter-dye distance changes between the different states (native, unfolded, and, if present, compact intermediate) were only observed if one of the dye attachment positions was close to a further secondary structure element (which was not part of the Rossmann fold, but which represented one or two β-strands that were also part of the enclosed structure elements (i.e., highlighted in color)). This condition was fulfilled for the N3-and C-3 variants (TS) and for the N1-and C2-variants (CI). Importantly, it was not fulfilled for the N2- and C1-variants (NT).Looking at the domain topologies, differences between the two-state and compact intermediate transitions seem to be related to which position within the Rossmann fold one of the two dye attachment sites was located in. A more upstream position of the dye attachment site appeared to reveal the compact intermediate, whereas a more downstream attachment site did not (position 1 versus position 34 for the N-variants and position 202 versus position 256 for the C-variants).

These results indicate that in the compact intermediate transition, one β-strand (for the C-domain 281–286) or two β-strands (for the N-domains 128–132 and 135–139) was at a non-native (i.e., too close) distance from the elements of the Rossmann fold. Thus, a wrong secondary structure packing seemed to occur transiently, which needed to be “unfolded” again before the correct structure could be formed (off-pathway intermediate), see for example [9,25]. 

## 4. Conclusions and Outlook

In the two-domain protein studied here, measuring six different intra-domain distances during the unfolding transition allowed the folding behavior of PGK to be studied in detail. Three distances each were measured in the N- and C-terminal domains. For all distances within a domain, different properties of the unfolding transitions were observed, which were able to be classified into three types of unfolding transitions. It was found that all three types were observed for each of the two domains. This result can certainly be explained by the large structural homology of the two domains. Our data suggest that a compact intermediate transiently appears in both domains during the folding/unfolding transition, but it is only visible in one of the variants (i.e., only for the corresponding intra-domain distance) for each domain. Unfortunately, only the six variants presented here could be produced for our smFRET analyses (for a detailed discussion of the selection, production, and limitations of suitable PGK variants, see [4]). Further and additional variants may provide an answer to the question of whether additional variants would have led to other types of folding transitions not yet observed. Previously published smFRET studies which mapped multiple distances in proteins with different domain topologies indicate that additional types of unfolding transitions that arise from equilibrium measurements can be found. In the case of the cytolytic toxin ClyA (helical monomer with 34 kDa), which forms an anti-parallel coiled-coil helical bundle, six different intermolecular distances were measured during GndHCl-induced unfolding. Five different types of unfolding transitions were revealed. In addition to a classical two-state transition, four transitions with intermediates (in two cases, also compact intermediates) were found. Again, the existence of intermediate states was only visible in the smFRET but not in the CD results [10]. In another example with the Acyl-CoA binding protein, also a protein with a helix-bundle topology but much smaller in size (10 kDa), four intramolecular distances were measured as functions of GndHCl. The corresponding smFRET data exhibited only two-state transitions without any observable intermediate state [8]. Finally, a study of apoflavodoxin (20 kDa) with an α-β topology quite similar to the domain topology of PGK revealed a compact intermediate unfolding transition type twice for two different intra-molecular distances [9]. The examples presented show that a particular unfolding type (e.g., compact intermediate) can indeed occur in proteins with different domain topologies. However, a greater diversity of unfolding types is more likely to occur in larger proteins. 

Another important aspect that has also not been addressed in this study is the extent to which the presence of neighboring domains (in the case of multi-domain proteins) has an impact on the folding properties of a selected domain. An answer to this question can be obtained from studies with isolate N- and C-terminal domains (truncated versions of yPGK). Comparison with the results from studies with full-length variants may then improve the understanding of how inter-domain interactions contribute to the folding of individual domains.

## Figures and Tables

**Figure 1 biomolecules-13-01280-f001:**
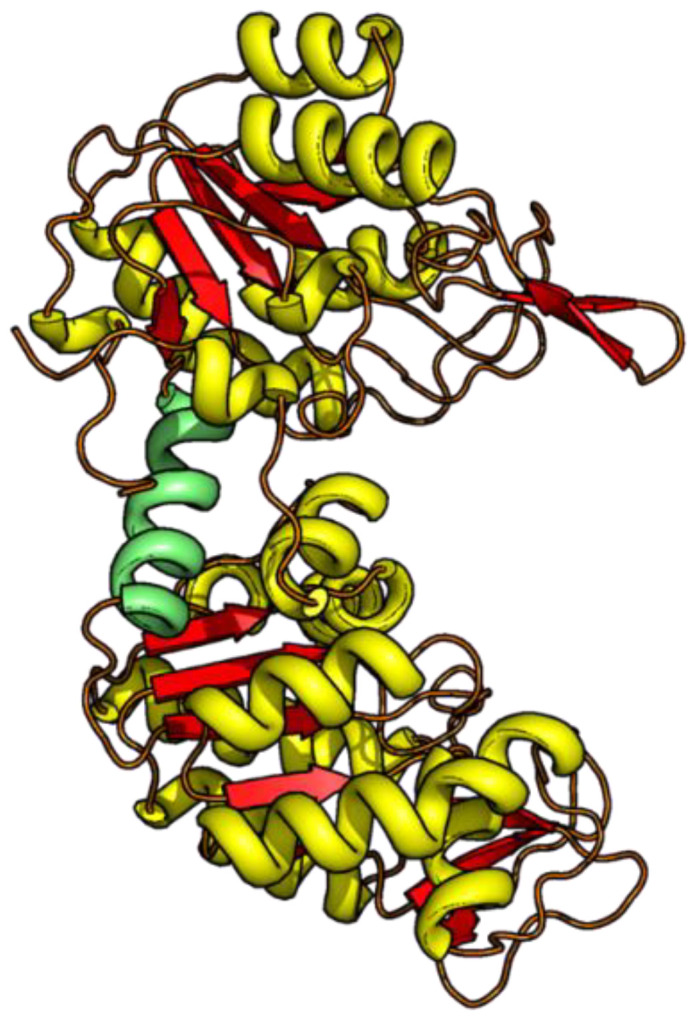
The 3D structure of yPGK (PDB: 1QPG) with the N-terminal domain (residues 1–185, upper domain) and the C-terminal domain (residues 205–415, lower domain) is depicted in the ribbon representation. The connecting hinge region (residues 186–204) is highlighted in green.

**Figure 2 biomolecules-13-01280-f002:**
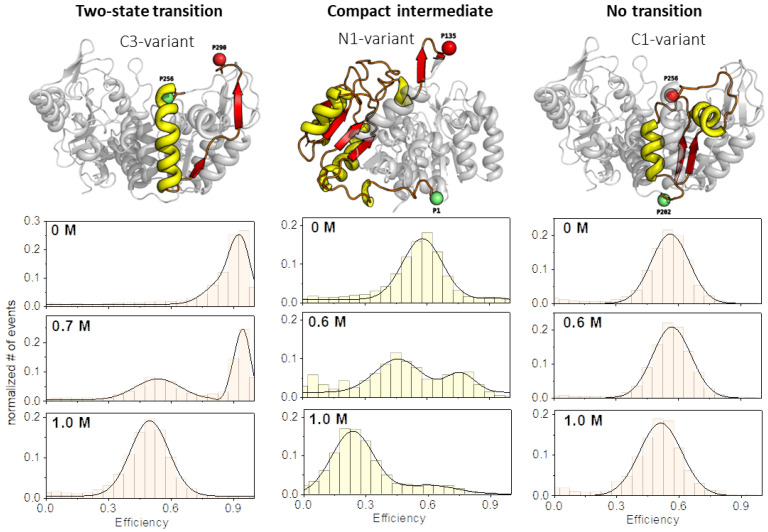
Examples of the three different unfolding transition types. The evolution of corresponding transfer efficiency histograms is shown as function of the denaturant concentration for the respective yPGK variant (from **top** to **bottom**: 0 M: native state; 0.6–0.7 M: half-transition regime; ≥1 M: fully unfolded state). The presented unfolding transitions can be classified in the following types: two-state transition (**left** column), a transition with a compact intermediate state (**middle** column), and a transition where the inter-dye distance does not change significantly (“no transition”) upon unfolding (**right** column).

**Figure 3 biomolecules-13-01280-f003:**
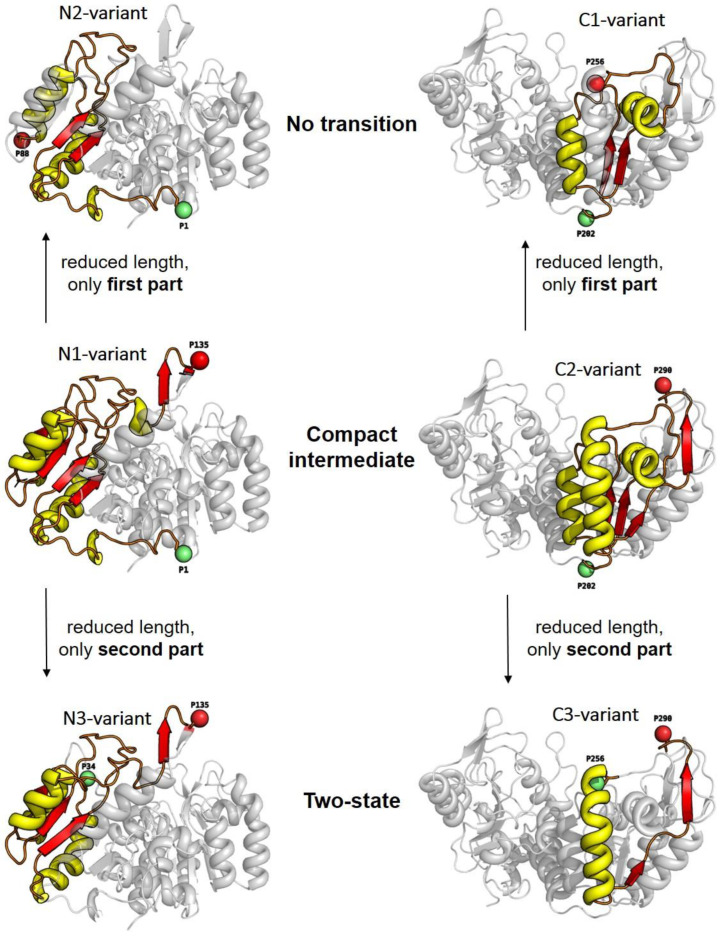
The 3D structures of yPGK are shown with individual attachment positions (red and green spheres) for all intra-domain distances within the N-terminal domain (**left** column) and within the C-terminal domain (**right** column). Structural elements which are enclosed between the dye attachment positions are highlighted with yellow (α-helices) and red (β-strands/sheets) colors. Relations between the number of residues between dye attachment positions and the position of the enclosed structural elements within the whole domain structure, on the one hand, and the corresponding types of unfolding transitions, on the other hand, are illustrated by the arrangement of the individual variants.

**Figure 4 biomolecules-13-01280-f004:**
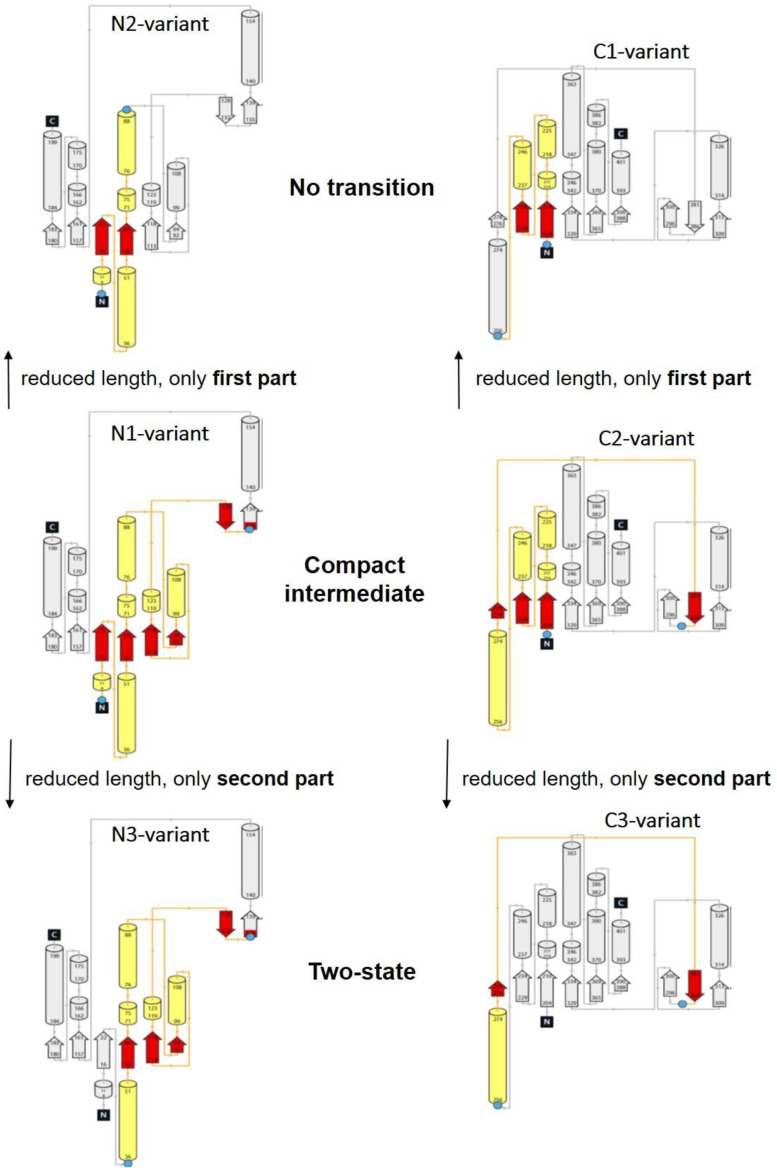
According to the protein structures shown in Figure 3, we show here the corresponding topology plots for the N-terminal domain (**left** column) and for the C-terminal domain (**right** column) for the same variants. For both domains, a Rossmann fold is recognizable. It is composed of six parallel beta strands that form an extended beta sheet. The strands are connected by α-helices resulting in a repeating α-β-α structure. Again, the structural elements between the dye attachment positions are highlighted with yellow (α-helix) and red (β-strands/sheets) colors.

**Table 1 biomolecules-13-01280-t001:** Table summarizing the results of all intra-domain distance changes, as obtained from smFRET histograms (see Appendix A). In addition to the number of residues between the dye attachment positions (# residues), the native, intermediate, and unfolded state inter-dye distances (R_DA_) are given. These values were obtained from the peak position of the corresponding population (at <*E*>) shown in the respective smFRET histograms. Following the idea that unfolded proteins exhibit the statistics of the end-to-end separation of a polymer chain composed of *n* amino acids (i.e., # residues), a spatial extension of the unfolded state was estimated, see text. The following types of transition were assigned: CI: compact intermediate; NT: no transition; TS: two-state.

	N-Domain			C-Domain		
Label Positions	1–135 N1-Variant	1–88 N2-Variant	34–135 N3-Variant	202–256 C1-Variant	202–290 C2-Variant	256–290 C3-Variant
# Residues	135	88	101	54	88	34
Native R_DA_ (0 M)	48 Å	51 Å	38 Å	51 Å	57 Å	31 Å
Intermediate R_DA_ (0.6–0.7 M)	43 Å	-	-	-	45 Å	-
Unfolded R_DA_ (≥1 M)	75 Å	52 Å	65 Å	51 Å	63 Å	54 Å
Polymer $\sqrt{\langle R^{2}\rangle }$	76 Å	61 Å	65 Å	48 Å	61 Å	38 Å
Type of transition	CI	NT	TS	NT	CI	TS

## Data Availability

Data which are contained within the article are available on request from the corresponding author.

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
