# Peer review of "Features of Protein Unfolding Transitions and Their Relation to Domain Topology Probed by Single-Molecule FRET"

_biomolecules, 2023, doi:10.3390/biom13091280_

Round 1

Reviewer 1 Report

Bustorff et al. employ single-molecule FRET to understand the unfolding dynamics of the PGK protein. To that aim, they designed six different Cys mutants to label their protein at different positions and investigate the topology of the unfolded state, in particular involving conformational changes in the homologous N and C terminal domains. The manuscript seems to be a continuation of a previous 2020 publication, here including more variants. However, reading the previous paper, it becomes hard to understand the novelty of the present manuscript (besides sampling different reaction coordinates); in other words, what new science do the authors learn in this study compared to the previous one? The authors should clarify the novelty of the present paper in contrast to the previous publication and what the added value here is. 

Another problem is the difficulty of following the results, all data are simply presented as histograms of the FRET efficiency, but it is not clear what reaction coordinates are sampled for each condition (for example, in Fig. 2). The authors should accompany these plots with relevant schemes of the labelled protein to help understand the reported states. Moreover, plotting the histograms as a function of the fluorophore distance (instead of the FRET efficiency) could also help in this direction.

Moreover, provided the work is a single-molecule study, I'm missing some single-molecule FRET trajectory that illustrates how the dynamics of the unfolding protein are. Since all that is shown are histograms, all the dynamic properties of the conformational transitions are lost. Can the authors extract any kinetic parameter from their experiments (e.g. folding/unfolding rates)? This would enrich the scientific meaningness of their work and also adequately extract all possible information from their experimental assay.

Finally, a critical assumption of their work is that the variants behave as the WT. I guess this is an intrinsic limitation of any FRET work. However, have the authors conducted any control to validate this assumption? How are the authors sure that, for example, the unfolding pathway through the compact intermediate is not an artifact of that particular variant? Can they perhaps conduct some bulk-like study to characterize the unfolding kinetics of that variant against the WT and demonstrate that the PGK protein unfolds through an intermediate?

English is ok. Some writing is confusing, and the conclusions are not too clear.

Reviewer 2 Report

The paper describes a single molecule FRET-based approach to monitor the Gnd.HCl-induced unfolding of yeast phosphoglycerate kinase (PGK). PGK is a 415-residue protein composed by two domains (N- and C-domain), exhibiting Rossmann-like a/b/a topologies. The work builds upon previous work based on the production of PGK variants designed to specifically map/pinpoint the unfolding process at different locations. The authors observe relevant changes in protein compactness that can be reporting the formation of unfolding intermediates. Results are displayed and described very nicely. The paper is well organized and well written and brings novelty to the field of protein unfolding. Even so, the manuscript can be improved by:

1. Major issues

1.1 Unfolding state

In order to corroborate the smFRET results and show that the unfolding state is indeed reached at 1.0 M of chaotropic denaturant (Gnd.HCl) it would be essential to add to the manuscript fluorescence emission spectra and/or Gnd.HCl-unfolding curves of fluorescence emission maxima versus [Gnd.HCl] (e.g.: from 0.0 to 6.0 M [Gnd.HCl]), to ensure that it is the unfolded state of the protein that is reached at 1.0 M Gnd.HCl, and it’s not only just a thermodynamically stable unfolding intermediate.

1.2 Additional data

Given the presence of the Rossmann motif and the typical profile of secondary structure in beta-sheet and alpha-helix that characterizes the protein, results would be more robust if far-UV circular dichroism spectra and/or fluorescence spectra would be added to enrich the manuscript, as a way to monitor the changes in secondary structure and domain topology happening during unfolding (that justify the changes in compactness observed by smFRET).

2. Minor issues – Typing issues

2.1 Introduction section – Line 57: Please replace “Recently, we performed already a study” by: “We have recently performed a study (…)”

2.2 Please separate values from units and use comma for thousands throughout the manuscript: e.g., 5,000 x g – (Line 85).

2.3 Please introduce under script in (NH4)2SO4 (lines 83 and 85).

2.4 Please include the extended form “guanidine hydrochloride” before using the GndHCl abbreviation (line 140).

2.5 Consider removing the adverb “mainly” (line 163) to promote clarity.

2.6 Please replace “with distances we obtain” by the past tense “with the distances we obtained” (line 165).

2.7 Please replace “Comact intermediate” by “Compact intermediate” (line 177).

2.8 Please use the past tense “chose” (line 181).

2.9 Please introduce the particle “the”: “the found folding schemes” (line 211).

2.10 Please replace “die” by “the” (line 218).

2.11 Please used the Greek letter alpha: a-helices (line 268).

2.12 Consider using a PyMol official reference – Ref.18 (e.g. Schrödinger, L., & DeLano, W. (2020). PyMOL. Retrieved from pymol.org/pymol).

Minor issues - Please correct the typing issues listed above.

Round 2

Reviewer 1 Report

The authors have now successfully addressed my queries, clarifying the novelty, and technical aspects, and including new diagrams that help in interpreting the data. I now recommend the manuscript for publication.

Reviewer 2 Report

The manuscript has been improved.